# REPRESENTING LATENT DIMENSIONS USING COMPRESSED NUMBER LINES

## ABSTRACT

Humans use log-compressed number lines to represent different quantities, including elapsed time, traveled distance, numerosity, sound frequency, etc. Inspired by recent cognitive science and computational neuroscience work, we developed a neural network that learns to construct log-compressed number lines. The network computes a discrete approximation of a real-domain Laplace transform using an RNN with analytically derived weights giving rise to a log-compressed timeline of the past. The network learns to extract latent variables from the input and uses them for global modulation of the recurrent weights turning a timeline into a number line over relevant dimensions. The number line representation greatly simplifies learning on a set of problems that require learning associations in different spaces – problems that humans can typically solve easily. This approach illustrates how combining deep learning with cognitive models can result in systems that learn to represent latent variables in a brain-like manner and exhibit human-like behavior manifested through Weber-Fechner law.

## 1 INTRODUCTION

The human ability to map sensory inputs onto number lines is critical for rapid learning, reasoning, and generalizing. Recordings of activity from individual neurons in mammalian brains suggest a particular form of representation that could give rise to mental number lines over different variables. For instance, the presentation of a salient stimulus to an animal triggers sequential activation of neurons called *time cells* which are characterized by temporally tuned unimodal basis functions (MacDonald et al., 2011; Tiganj et al., 2017; Eichenbaum, 2014). Each time cell reaches its peak activity at a particular time after the onset of some salient stimulus. Together, a population of time cells constitutes a temporal number line or a timeline of the stimulus history (Howard et al., 2015; Tiganj et al., 2018). Similarly, as animals navigate spatial environments neurons called *place cells* exhibit spatially tuned unimodal basis functions (Moser et al., 2008). A population of place cells constitutes a spatial number line that can be used for navigation (Bures et al., 1997; Banino et al., 2018). The same computational strategy seems to be used to represent other variables as well, including numerosity (Nieder & Miller, 2003), integrated evidence (Morcos & Harvey, 2016), pitch of tones (Aronov et al., 2017), and conjunctions of these variables (Nieh et al., 2021). Critically, many of these "neural number lines" appear to be log-compressed (Cao et al., 2021; Nieder & Miller, 2003), providing a natural account of the Weber-Fechner law observed in psychophysics (Chater & Brown, 2008; Fechner, 1860/1912). Here we present a method by which deep neural networks can construct continuous, log-compressed number lines of latent task-relevant dimensions.

Modern deep neural networks are excellent function approximators that learn in a distributed manner: weights are adjusted individually for each neuron. Neural activity in the brain suggests a representation where a population of neurons together encodes a distribution over a latent variable in the form of a number line. In other words, a latent variable is not represented as a scalar (*e.g.*, a count of objects could be encoded with a single neuron with a firing rate proportional to the count), but as a function supported by a population of neurons, each tuned to a particular magnitude of the latent variable. To build deep neural networks with this property, we use global modulation such that recurrent weights of a population of cells are adjusted simultaneously. We show that this gives rise to the log-compressed number lines and can greatly facilitate associative learning in the latent space.

Inspired by experiments on animals, we conduct experiments where a neural network learns number lines for spatial distance and count of objects appearing over time. We designed an experimental setup where the network needs to predict when a target event will happen. In our experiments, time to the target event depends either on the elapsed time, traveled spatial distance, or the count of how many times some object appears in the input. Critically, just like in the experiments with animals, these variables are not directly observable from the inputs – they are hidden and have to be learned from the spatiotemporal dynamics of the input. For example, if the target event will happen after some object appears a certain number of times, the network needs to learn to identify the object and learn the correct number of appearances (Experiment 3; see also an illustration in Fig. 5). Similarly, people can estimate distance when riding a bicycle using their motor outputs and sensory inputs – we can learn a non-linear mapping of motor outputs and sensory inputs onto velocity and integrate velocity to estimate distance (this concept is an inspiration for Experiments 2a and 2b; see also an illustration in Fig. 4).

Building on models from computational and cognitive neuroscience (Shankar & Howard, 2012; Howard et al., 2014; 2018), we propose a neural network architecture that gives rise to a number line supported by unimodal basis functions. The network is composed of three layers. The input is fed into a recurrent layer with the weights analytically computed to approximate the real-domain Laplace transform of the input. Critically, we use properties of the Laplace domain and apply global modulation to the recurrent weights to convert functions of time into functions of other variables, such as distance or count. The output of the Laplace layer is mapped through a linear layer with analytically computed weights implementing the inverse Laplace transform. The inverse gives rise to a log-compressed number line supported with unimodal basis functions. Depending on the modulatory signal, this population can resemble, for instance, time cells, place cells or count cells. The output is then mapped through a trainable dense layer with a sigmoid activation function to the network's output. This approach augments the capabilities of cognitive models, which typically rely on handcrafted features enabling them to learn latent variables. At the same time, the structure and properties of the cognitive model are preserved, allowing the resulting system to have strong explanatory power of neural activity in the brain and behavioral data.

To the best of our knowledge, this work is the first time that the Laplace transform and its inverse have been implemented as a part of a neural network trainable with error backpropagation – in our experiments, the gradient flows through the Laplace and inverse Laplace transform. Global modulation of analytically computed recurrent weights is also a novel approach to learning in Recurrent Neural Networks (RNNs). The Laplace transform uses a diagonal weight matrix which makes it robust to problems of exploding and vanishing gradients related to backpropagation through time. These problems have been somewhat reduced with gated networks, such as Long Short-Term Memory (LSTM) (Greff et al., 2016; Hochreiter & Schmidhuber, 1997) and Gated Recurrent Units (GRU) (Chung et al., 2014). Recent approaches bounded the gradients, such as Coupled Oscillatory RNN (coRNN) (Rusch & Mishra, 2020) or used formalism that does not require learning of the recurrent weights, such as Lagrange Memory Unit (LMU) (Voelker et al., 2019). Echo state networks (Jaeger, 2001) used RNNs with non-trainable, fixed recurrent weights but without global modulation. Similarly, the multiscale temporal structure approach used fixed weights with a spectrum of time constants (Mozer, 1992).

## 2 METHODS

We first describe the construction of a log-compressed timeline using an approximation of the real-domain Laplace and the inverse Laplace transform. Then we describe how the timeline can be turned into a more general number line via global weight-modulation in the Laplace domain.

### 2.1 CONTINUOUS-TIME FORMULATION OF LAPLACE AND INVERSE LAPLACE TRANSFORM

Given a one-dimensional input signal $f(t)$, we define a modified version of the Laplace transform $F(s; t)$:

$$F(s; t) = \int_0^t e^{-s(t-t')} f(t')dt'. \tag{1}$$

This modified version differs from the standard Laplace transform only in variable $s$: instead of $s$ being a complex value composed of the real and imaginary part, we use real and positive $s$. This

modification simplifies the neural network implementation while still giving us the computational benefits of the standard Laplace transform, as we will illustrate below. Note that $F$ is also a function of time $t$. This implies that at every moment we construct the Laplace transform of the input function up to time $t$: $f(0 \leq t' < t) \xrightarrow{L} F(s;t)$.

To construct the timeline we need to invert the Laplace transform. The inverse which we denote as $\tilde{f}(\overset{*}{\tau};t)$ can be computed using Post inversion formula (Post, 1930):

$$\tilde{f}(\overset{*}{\tau};t) = \mathbf{L}_k^{-1} F(s;t) = \frac{(-1)^k}{k!} s^{k+1} \frac{d^k}{ds^k} F(s;t), \tag{2}$$

where $\overset{*}{\tau} := k/s$ and $k \to \infty$.

## 2.2 GENERALIZING A TIMELINE INTO A NUMBER LINE

We can use the Laplace domain to convert a timeline into a number line if we learn a time derivative of a variable that we want to represent using the number line. To achieve this, instead of computing the Laplace transform for a function of time $f(t)$, we compute the transform for a function of some variable $f(x(t))$.

We first rewrite Eq. 1 in a differential form:

$$\frac{dF(s;t)}{dt} = -sF(s;t) + f(t). \tag{3}$$

The impulse response (response to input $f(t) = \delta(0)$) of $F(s;t)$ decays exponentially as a function of time $t$ with decay rate $s$: $e^{-st}$ (Fig. 1a).

We define a time derivative of $x(t)$ as $\alpha(t) = \frac{dx}{dt}$ and modify Eq. equation 3 as follows:

$$\frac{dF(s;t)}{dt} = -\alpha(t)sF(s;t) + f(t), \tag{4}$$

such that $\alpha(t)$ modulates the decay rate $s$ (Fig. 1b).

By reorganizing terms in Eq. equation 4 and applying the chain rule we obtain the Laplace transform of $f(x(t))$:

$$\frac{dF(s;x)}{dx} = -sF(s;x) + f(x). \tag{5}$$

Note that Eq. equation 3 and Eq. equation 5 are the same, except that $t$ in Eq. equation 3 is $x$ in Eq. equation 5. This allows us to use the same calculation as in Eq. equation 2 to obtain the inverse Laplace transform of $\tilde{f}(\overset{*}{\tau};x)$.

## 2.3 NEURAL NETWORKS IMPLEMENTATION OF THE LAPLACE AND INVERSE LAPLACE TRANSFORM WITH GLOBAL WEIGHT MODULATION

We implement an approximation of the modified Laplace and inverse Laplace transform as a two-layer neural network with analytically computed weights (Fig. 1c). The first layer implements the modified Laplace transform through an RNN with recurrent weights. The second layer implements the inverse Laplace transform as a dense layer with weights analytically computed to implement $k - th$ order derivative with respect to $s$.

While in the Laplace domain $s$ is a continuous variable, here we redefine $s$ as $N$ elements long vector. We can now write a discrete-time approximation of Eq. equation 4 as an RNN with a diagonal connectivity matrix and a linear activation function:

$$F_{s;t} = W_L F_{s;t-1} + f_t, \tag{6}$$

where $W_L$ is a recurrent weight matrix $W_L = e^{-\alpha(t)S\Delta t}$ where $W_L$ is an $N$ by $N$ matrix and $S$ is a diagonal matrix with diagonal elements composed of $s$ values. At every time step $t$, $F_{s;t}$ is $N$ elements long vector. For brevity of the notations, we assume that the duration of a discrete-time step $\Delta t = 1$. Importantly, since the values of vector $s$ are fixed, $W_L$ is also fixed and not learned during training.

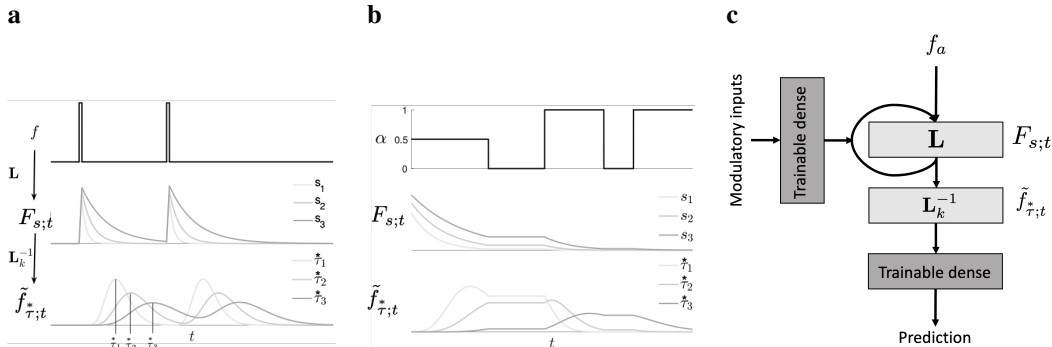

Figure 1: The activity of neurons in CNL with and without modulatory input. **a.** Impulse response of the Laplace transform approximation decays exponentially with decay rate $s$. The impulse response of the inverse Laplace transform approximation has a unimodal form such that each curve peaks at $\overset{*}{\tau}$. Note that if time $t$ was shown on the log-scale, the unimodal curves would be equally wide and equidistant. **b.** $\alpha$ modulates the decay rate of $F_{s;t}$. When $\alpha = 0$, $F_{s;t}$ is constant since its time derivative is 0. When $\alpha = 0.5$ the decay of $F_{s;t}$ is slowed down by a factor of 2. **c.** Schematic of the CNL network with input $f_a$ and modulatory inputs. $f_a$ feeds into the recurrent layer that implements an approximation of the Laplace transform using analytically computed recurrent weights. Modulatory inputs are passed through a trainable dense layer and modulate the recurrent weights. Approximation of the inverse Laplace transform $\tilde{f}_{\overset{*}{\tau};s}$ is computed by applying a linear operator $L_k^{-1}$ on $F_{s;t}$. Afterward, $\tilde{f}_{\overset{*}{\tau};s}$ is fed into a trainable dense layer with sigmoid activation function. The network has a single prediction output which is trained to predict the activity of target $f_b$.

Following Eq. equation 2, a discrete approximation of the inverse Laplace transform, $\tilde{f}_{\overset{*}{\tau};t}$, can be implemented as a dense layer on top of $F_{s;t}$. The connectivity matrix of the dense layer is $\mathbf{L}_k^{-1}$(see Appendix Sec. A.1 for the derivation of the exact matrix form of $\mathbf{L}_k^{-1}$).

To interpret $\tilde{f}_{\overset{*}{\tau};t}$ and to select values $s$ in an informed way, we compute the impulse response of $\tilde{f}_{\overset{*}{\tau};t}$. For input $f(t) = \delta(0)$, the activity of $\tilde{f}_{\overset{*}{\tau};t}$ is:

$$\tilde{f}_{\overset{*}{\tau};t} = \frac{1}{u(t)} \frac{k^{k+1}}{k!} \left(\frac{u(t)}{\overset{*}{\tau}}\right)^{k+1} e^{-k\frac{u(t)}{\overset{*}{\tau}}}, \tag{7}$$

where $u(t = t_n) = \sum_{i=0}^{t_n} \alpha(t_i)$. If $\alpha(0 \leq t < t_n) = 1$, then $u(t = t_n) = t_n$. To investigate properties of this approximation and to motivate our choice for $\overset{*}{\tau}$ and consequently $s$, we analyze the case where $\alpha(0 \leq t \leq t_n) = 1$.

The impulse responses of units in $\tilde{f}_{\overset{*}{\tau};t}$ is a set of unimodal basis functions (Fig. 1a and Fig. 2a). To better characterize its properties we first find the peak time by taking a partial derivative with respect to $t$, equate it with 0 and solve for $t$: $\partial\tilde{f}_{\overset{*}{\tau};t}/\partial t = 0 \rightarrow t = \overset{*}{\tau}$. Therefore each unit in $\tilde{f}_{\overset{*}{\tau};t}$ peaks at $\overset{*}{\tau}$. Note that if we computed the exact continuous-time Laplace and inverse Laplace transform (which would require infinitely many neurons, since $s$ and $\overset{*}{\tau}$ would be continuous variables), the impulse response would be a $\delta(\overset{*}{\tau})$. This would provide a perfect reconstruction of the input function $f(0 < t' < t)$ rather than its approximation.

To further characterize our approximation, we express the width of the unimodal basis functions of the impulse response of $\tilde{f}_{\overset{*}{\tau};t}$ through the coefficient of variation $c$ (see Appendix Sec. A.2 for the derivation of $c$): $c = 1/\sqrt{k+1}$. Importantly, $c$ does not depend on $t$ and $\overset{*}{\tau}$, implying that the width of the unimodal basis functions increases linearly with their peak time. Therefore when observed as a function of $\log(t)$, the width of the unimodal basis functions is constant. Note that this property is critical for log-compression.

We choose values of $\overset{*}{\tau}$ as log-spaced between some minimum and maximum (see next section for the values used in the experiments). Because of the log-spacing and because c does not depend on $t$ and $\overset{*}{\tau}$, when analyzed as a function of $log(t)$, the unimodal basis functions are equidistant and equally wide, providing uniform support over $log(t)$ axis. This result is analogous to the scale-invariance observed in human timing and perception, formalized as the Weber-Fechner law. Intuitively, this is beneficial since the more recent past carries more predictive power than the more distant past. Hence, our approximation of function $f(0 < t' < t)$ will be better for values closer to $t$ than to 0. Note that fixing the values of $\overset{*}{\tau}$ and choosing $k$ also fixes values of $s$ since $s = k/\overset{*}{\tau}$ so $s$ is not a trainable parameter.

To convert a log-compressed timeline into a log-compressed number line, we implemented global modulation of recurrent weights as in Eq. equation 4. We assume that $\alpha$ is not known and that it needs to be learned from the input data. $\alpha(t)$ is computed through a feedforward projection which receives a subset of inputs designated as modulatory inputs (*e.g.*, inputs from which a latent variable, such as velocity or count, can be computed). Details of the feedforward projection are stated in the description of each experiment since they varied across experiments due to variability in the input dimensionality. We feed $\tilde{f}_{\overset{*}{\tau};t}$ into a trainable dense layer with a single output and a sigmoid activation function. Due to its log-compression of the input signal and gating of the recurrent weights, we call our approach Compressed Number Line or CNL.

## 3    RESULTS

We tested the proposed network on four event prediction experiments inspired by psychophysics experiments in humans and other animals. In our experiments, event $a$ predicted event $b$ such that the time between them depends on either the actual amount of elapsed time (Experiment 1), traveled distance with velocity as a latent variable which can be learned as a non-linear combination of the inputs (Experiments 2a and 2b) or the number of occurrences of a specific pattern (Experiment 3). Our goal was to demonstrate that the proposed network can identify latent variables by learning an appropriate mapping from inputs onto a global modulatory signal $\alpha$. The network then constructs a number line representation over an integral of the latent variable (*e.g.*, if velocity is a latent variable, then traveled distance is its integral across time), making the learning more accurate, faster, and more robust than with other approaches.

CNL parameters were kept the same for all the experiments. We set $k = 8$ and we used 64 neurons in $F$ and 50 neurons in $\tilde{f}$ with input $f_a$. The number of neurons in $\tilde{f}$ was smaller than in $F$ in order to avoid edge effects when taking a derivative with respect to $s$ (the number of units in $\tilde{f}$ is $2k$ less than the number of units in $F$; see Appendix Sec. A.1 for details on the computation of the discrete approximation of the derivative). $\overset{*}{\tau}$ was composed of 50 log-spaced values between 5 and 20000. These lower and upper limits were chosen to cover the entire range of temporal relationships in the input signal. The output of $\tilde{f}$ layer was fed through trainable weights into a single output neuron with a sigmoid activation function.

To evaluate how commonly used RNNs perform on this task we compared CNL with a simple RNN, GRU, LSTM, LMU, and coRNN followed by a fully connected layer. For LMU we experimented with hyperparameters and got the best performance with: $dt = 1$, $\theta = 5000$ and $d = 128$. For coRNN we set the hyperparameters to the same values as in Rusch & Mishra (2020): $dt = 0.016$, $\gamma = 94.5$ and $\epsilon = 9.5$. The hidden size was 64 in all cases. As with CNL, the output layer had a single neuron with a sigmoid activation function. While still relatively simple, these networks had many more parameters than CNL (*e.g.*, Table 1) because most of the CNL parameters were computed analytically.

The results were obtained as an average of three runs with 95% confidence intervals. Each of the networks was trained for 1000 epochs using the learning rates 0.001, 0.01, 0.1, and 1.0 with ADAM optimization on an Nvidia Tesla P100 12GB GPU. We used sample weighting to train the networks using binary cross-entropy as supervising loss and report binary cross-entropy as test loss. We also report the average distance between the actual timestamp of the target event and the timestamp that received the highest probability estimate by a particular network.

### 3.1 EXPERIMENT 1: PREDICTING INTERVAL DURATION AT A WIDE RANGE OF TEMPORAL SCALES

To test the baseline performance of different approaches, we start with a simple experiment with no latent variables. We designed a setup with input $f_a(t) = \delta(x)$ and target (label) $f_b(t) = \delta(x + x')$ (Fig. 2a). We used three values for $x'$: $x' \in [50, 500, 5000]$. Length of $f_a(t)$ and $f_b(t)$ was $4x'$, 200, 2000 and 20000. We created only 3 training examples and 12 validation and 35 test examples, every time with a random $x$ sampled uniformly from a range between 1 and $3x'$.

Results are shown in Table 1 (cross-entropy loss) and Table 5 (average distance). At the shortest timescale with $x' = 50$ and medium timescale with $x' = 500$, LMU did a perfect job of predicting $f_b$. At the largest time scale ($x' = 5000$), CNL performed better than the other approaches, which effectively failed to provide a useful prediction (Fig. 6), likely due to a large number of parameters and problems with gradient backpropagation in long temporal sequences.

Table 5 indicates that for CNL the error in the distance grows roughly linearly with $x'$. This gradual drop in performance is due to the log-spacing of the recurrent weights. Fig. 6 shows how the standard deviation of the bell-shaped prediction curve scales with the temporal distance. These properties are consistent with temporal estimates in humans and the Weber-Fechner law.

To investigate the importance of the inverse Laplace transform, we also compared CNL (which is composed of the Laplace and inverse Laplace transform) with $F(s)$ (only the Laplace transform, without the inverse) combined with a trainable dense layer of 50 neurons. We label the later model as CNL-F. CNL performed better than CNL-F at all scales. We attribute this to the fact that the inverse Laplace transform constructs a timeline, which is a representation that makes learning temporal associations particularly easy. Timeline representation means that neurons activate sequentially following event $a$. This means that some neurons are active only when event $b$ happens, making it easy for the network to learn the temporal relationships. This is confirmed by observing the learned weights of the dense layer in CNL (Fig. 3). Since the Laplace transform is linear, the output weights are interpretable. The weights are largest around $\overset{*}{\tau} = x'$ and have a Mexican hat connectivity pattern which helps to sharpen the prediction peak (Fig. 6).

|  | x'=50 | x'=500 | x'=5000 | Params |
|---|---|---|---|---|
| CNL-F | $0.399 \pm 0.031$ | $0.433 \pm 0.008$ | $0.639 \pm 0.014$ | **51** |
| CNL | $0.212 \pm 0.003$ | $0.227 \pm 0.004$ | $\mathbf{0.255 \pm 0.004}$ | **51** |
| RNN | $0.521 \pm 0.317$ | $0.446 \pm 0.281$ | $0.561 \pm 0.184$ | 4353 |
| LSTM | $0.347 \pm 0.500$ | $0.669 \pm 0.077$ | $0.647 \pm 0.000$ | 17217 |
| GRU | $0.103 \pm 0.111$ | $0.326 \pm 0.559$ | $0.646 \pm 0.001$ | 12929 |
| coRNN | $0.264 \pm 0.255$ | $0.408 \pm 0.304$ | $0.647 \pm 0.000$ | 8385 |
| LMU | $\mathbf{0.005 \pm 0.008}$ | $\mathbf{0.208 \pm 0.265}$ | $0.732 \pm 0.103$ | 12610 |

Table 1: Experiment 1: Binary cross-entropy loss across different scales.

### 3.2 EXPERIMENT 2A: PREDICTING DISTANCE IN THE PRESENCE OF TWO MODULATORY INPUTS

In this experiment input $f_a$ predicts input $f_b$, but the time between them was modulated by other inputs, $f_c$ and $f_d$ (see Fig. 4 for an illustration of the experiment). This means that the effective duration of each time step was a weighted sum of $f_c$ and $f_d$ as described in Alg. 1. For the CNL network, this meant that it had to learn that $\alpha = w_1 f_c + w_2 f_d$, where $w_1$ and $w_2$ were randomly generated weights between 0 and 1 and $f_c$ and $f_d$ had a magnitude ranging from 0 to 5. The network only observed $f_c$ and $f_d$ and not the weights $w_1$ and $w_2$, so it had to learn the relationship between $\alpha$ and $f_c$ and $f_d$. We can view this problem as one of learning spatial distance between event $a$ and event $b$ such that velocity (in this case $\alpha(t)$) is not directly observable but has to be learned from the modulatory inputs $f_c$ and $f_d$. Therefore, $a$ does not predict $b$ after a particular amount of time has elapsed, but instead after traveling a particular distance at a velocity that has to be learned from the network inputs. Similar to Experiment 1, in the absence of modulatory inputs $f_a(t) = \delta(x)$ and $f_b(t) = \delta(x + x')$ with $x' \in [50, 500, 5000]$. Modulatory inputs $f_c$ and $f_d$ were stepwise functions broken into 15 pieces with amplitude of each piece between 0 and 10. The two modulatory inputs

were passed through a set of weights $W_\alpha$ (one weight for each modulatory input), adding two more parameters to CNL.

---

**Algorithm 1** Generating an interval duration $x'_m$ with modulatory inputs $f_c$ and $f_d$. If $f_c$ and $f_d$ are equal to 1, $x'_m = x' = 100$.

---

$i \Leftarrow 0$
$x' \Leftarrow 0$
**while** $i \leq x'$ **do**
    $i = i + 1$
    $x'_m = x'_m + w_1 f_c(x + i) + w_2 f_d(x + i)$
**end while**

---

We constructed 6 training, 22 validation, and 22 test examples. The error was again quantified as binary cross-entropy loss in predicting $f_b(t)$. Results from Experiment 2 are given in Table 2 (cross-entropy loss) and Table 6 (average distance). Despite having much fewer parameters, CNL performed better than the other approaches at all timescales. We attribute this to the ability of CNL to learn $\alpha$ from the inputs and cast the problem to be similar to Experiment 1 by constructing a number line and then simply learning which component on the number line corresponds to the target, which is a simple associative task. From observing plots in Fig. 7 we see that other approaches failed to provide a useful prediction.

|  | x'=50 | x'=500 | x'=5000 | Params |
|---|---|---|---|---|
| CNL-F | $0.537 \pm 0.171$ | $0.585 \pm 0.006$ | $0.633 \pm 0.047$ | **53** |
| CNL | $\mathbf{0.275 \pm 0.007}$ | $\mathbf{0.277 \pm 0.004}$ | $\mathbf{0.336 \pm 0.018}$ | **53** |
| RNN | $0.578 \pm 0.489$ | $0.574 \pm 0.316$ | $0.575 \pm 0.157$ | 4481 |
| LSTM | $0.637 \pm 0.349$ | $0.677 \pm 0.179$ | $0.617 \pm 0.179$ | 17729 |
| GRU | $0.653 \pm 0.098$ | $0.659 \pm 0.131$ | $0.594 \pm 0.096$ | 13313 |
| coRNN | $0.611 \pm 0.093$ | $0.593 \pm 0.071$ | $0.509 \pm 0.019$ | 8513 |
| LMU | $0.645 \pm 0.161$ | $0.592 \pm 0.258$ | $0.521 \pm 0.350$ | 12740 |

Table 2: Experiment 2a: Binary cross-entropy loss across different scales.

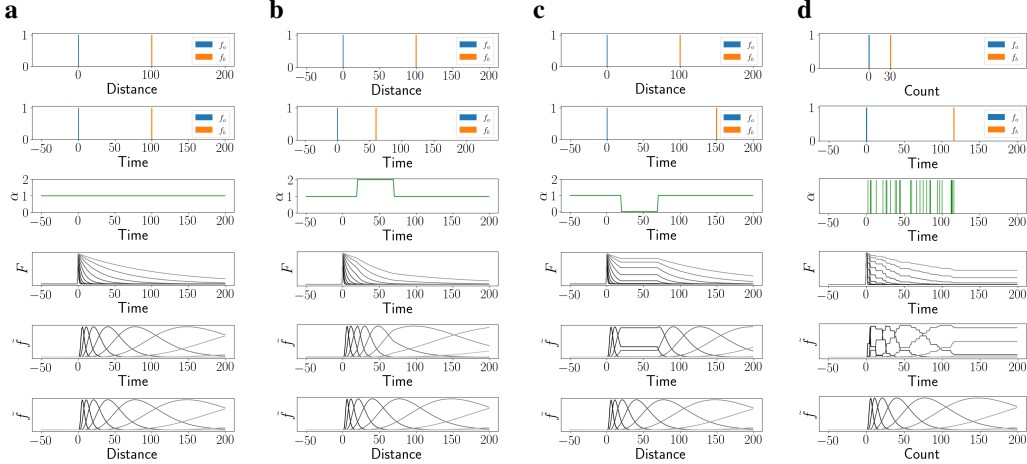

Figure 2: Temporal relationships in the presence of modulatory inputs. Delta pulse in $f_a$ predicts a delta pulse in $f_b$. **a** No temporal modulation (*e.g.*, $\alpha = 1$). **b** and **c** When the modulatory signal corresponds to velocity (Experiment 2), the network represents traveled distance as manifested by units that activate sequentially as a function of distance (bottom row). **d** Temporal modulation with delta pulses (Experiment 3). The network represents the total count of presented pulses.

|  | x'=50 | | x'=500 | | Params |
|---|---|---|---|---|---|
|  | Loss | Distance | Loss | Distance |  |
| CNL-F | $0.601 \pm 0.220$ | $17.4 \pm 7.6$ | $0.498 \pm 0.004$ | $229.7 \pm 0.0$ | **1352** |
| CNL | $\mathbf{0.2031 \pm 0.019}$ | $\mathbf{7.4 \pm 0.6}$ | $\mathbf{0.2601 \pm 0.020}$ | $\mathbf{52.6 \pm 8.8}$ | **1352** |
| RNN | $0.658 \pm 0.162$ | $70.4 \pm 15.9$ | $0.676 \pm 0.229$ | $746.6 \pm 96.0$ | 7553 |
| LSTM | $0.627 \pm 0.064$ | $67.3 \pm 3.7$ | $0.657 \pm 0.170$ | $675.4 \pm 224.0$ | 30017 |
| GRU | $0.578 \pm 0.134$ | $57.1 \pm 16.6$ | $0.640 \pm 0.075$ | $573.9 \pm 22.2$ | 22529 |
| coRNN | $0.627 \pm 0.196$ | $54.8 \pm 21.9$ | $0.627 \pm 0.043$ | $733.1 \pm 112.4$ | 11585 |
| LMU | $0.614 \pm 0.062$ | $104.3 \pm 2.7$ | $0.639 \pm 0.120$ | $561.3 \pm 239.0$ | 15860 |

Table 3: Experiment 2b: Binary cross-entropy loss across different scales and the average distance between the actual timestamp of the target event and the timestamp that received the highest probability estimate.

|  | count=10 | | count=200 | | Params |
|---|---|---|---|---|---|
|  | Loss | Distance | Loss | Distance |  |
| CNL-F | $0.469 \pm 0.249$ | $24.2 \pm 53.1$ | $0.085 \pm 0.001$ | $1005.0 \pm 0.0$ | **196** |
| CNL | $\mathbf{0.201 \pm 0.036}$ | $\mathbf{1.3 \pm 0.4}$ | $\mathbf{0.077 \pm 0.004}$ | $\mathbf{75.8 \pm 21.5}$ | **196** |
| RNN | $0.323 \pm 0.076$ | $69.8 \pm 24.5$ | $0.108 \pm 0.023$ | $1365.1 \pm 421.1$ | 5377 |
| LSTM | $0.303 \pm 0.140$ | $62.0 \pm 14.2$ | $0.096 \pm 0.010$ | $1264.4 \pm 941.1$ | 21313 |
| GRU | $0.324 \pm 0.119$ | $28.8 \pm 1.7$ | $0.116 \pm 0.072$ | $629.0 \pm 1006.0$ | 16001 |
| coRNN | $0.636 \pm 0.117$ | $60.8 \pm 25.3$ | $0.093 \pm 0.000$ | $1619.3 \pm 0.0$ | 9409 |
| LMU | $0.402 \pm 0.297$ | $65.5 \pm 87.8$ | $0.146 \pm 0.135$ | $597.7 \pm 781.4$ | 13650 |

Table 4: Experiment 3: Binary cross-entropy loss across different scales and the average distance between the actual timestamp of the target event and the timestamp that received the highest probability estimate.

### 3.3 EXPERIMENT 2B: PREDICTING DISTANCE IN THE PRESENCE OF FIFTY MODULATORY INPUTS

This experiment is a more complex version of Experiment 2a with fifty modulatory inputs instead of just two. In this case $\alpha = \sum_{i=1}^{50} w_i f_i$, where $f_i$ were the modulatory inputs and weights $w_i$ for $1 \leq i \leq 6$ were chosen randomly between 0 and 1 and weights $w_i$ for $7 \leq i \leq 50$ were 0. The range of values for the magnitude of $f_i$ was between 0 and 5. Thus the network had to learn to ignore 44 of the inputs (which were effectively noise) and add together a weighted sum of 6 inputs. If the network correctly learns $\alpha$, then the distance between $f_a$ and $f_b$ becomes fixed on a number line. The fifty modulatory inputs were passed through a set of weights $W_\alpha$, which was a two-layer feedforward network.

We constructed 60 training, 20 validation, and 20 test examples. We conducted the experiment at two temporal scales with $x' = 50$ and $x' = 500$. Since this experiment was more demanding in terms of resources, we omitted the timescale $x' = 5000$. Results from Experiment 2b are shown in Table 3. CNL again performed better than the other approaches at all timescales. Results shown in Fig. 8 suggest that CNL was again the only approach that provided a useful estimate.

### 3.4 EXPERIMENT 3: COUNTING

In Experiment 3, the time between $a$ and $b$ was modulated by the number of specific patterns presented as modulatory inputs (Fig. 5). At each time step, a pattern composed of 16 elements with binary values was presented. One of the patterns, which we refer to as the target pattern, was repeated multiple times and the network had to learn that the time between $a$ and $b$ depends on the number of target patterns. The patterns were generated randomly.

To solve this task, the network had to learn to recognize the target pattern and count its repetitions. In this case, $\alpha$ had to be learned as a temporal derivative of the count. Whenever the target pattern appears, the count changes by one, therefore $\alpha = 1$. Whenever the pattern was not the target pattern,

the counts stay the same and $\alpha = 0$. Fig. 2d shows an illustration of this, assuming that the network has learned appropriate $\alpha$.

We conducted the experiment at two different scales, one with 10 target counts and one with 200 target counts. In the first case, the input signal had 200 time steps, and in the second case, it had 2000 time steps. The results are shown in Table 4. At both scales, CNL performed better than the other approaches demonstrating that it learned the temporal derivative and constructed a number line representing the count of the target pattern. Fig. 9 shows prediction results for each of the seven networks.

## 4 DISCUSSION

The mammalian brain seems to commit to representing many physical and abstract dimensions as number lines. Computational neuroscience and cognitive science work developed a framework for the real-value Laplace transform and inverse Laplace transform that gives rise to such representation. Our approach uses that framework as a part of a deep neural network that can extract relevant latent dimensions. Because of commitment to the number line representation, the network can learn various associative relationships with very few training examples and a small number of weights. Existing widely-used networks, such as LSTMs and GRUs fail at most learning tasks used here. While, in principle, neural networks with multiple layers, many training examples and long training time could potentially perform well, the proposed approach provides a much simpler alternative.

Cognitive models provide immense utility in understanding neural computations in the brain, but they are usually limited to handcrafted features and associative learning. Here we have shown that incorporating cognitive models into neural networks can expand the utility and explanatory power of these models. CNL can take advantage of deep learning and learn latent features while at the same time utilizing the benefits of the cognitive model and structured representation of knowledge which allows easy associative learning. The resulting performance resembled human performance in psychophysics tasks, commonly characterized by the Weber-Fechner law. This is manifested by a roughly linear increase of error with distance in Table 5 and Table 6 as well as a linear increase of the standard deviation of the bell-shaped prediction curve in Fig. 6 as a function of distance.

This work introduces a trainable neural network implementation of real-domain Laplace transform and inverse Laplace transform. The Laplace domain enables access to a number of useful operations. For instance, translation, convolution and cross-correlation can be efficiently implemented in the Laplace domain. In this work, the network constructs the real-domain Laplace transform of the position of A as a function of the latent dimension. In the experiments used here, this is simply a delta function. Convolution of two delta functions at two real numbers produces a delta function at their sum $\delta(x) * \delta(y) = \delta(x + y)$; cross-correlation is similarly understandable as subtraction. In this way, abstract dimensions could be used with sparse representations as part of a number system for symbolic computation (Howard et al., 2015; Howard & Hasselmo, 2020). Because the same representational scheme can be generated for many different latent dimensions, the same operations could be reused for many different kinds of information. Supplementary material contains code and detailed instructions for reproducing the results and integrating the Laplace and inverse Laplace transform into deep neural networks.

Activity patterns of neurons in the proposed network resemble the activity of neurons recorded in mammalian brains. Fig. 2 shows time cells, place cells (in 1D environment) and cells tuned to a particular number of objects. Each of these cell types has been recorded in mammalian brains (Bures et al., 1997; MacDonald et al., 2011; Morcos & Harvey, 2016). The experiments used here require the network to learn a variety of complicated one-dimensional latent dimensions. Neurons in the mammalian hippocampus and prefrontal cortex have been shown to form conjunctive representations of multiple variables (Rigotti et al., 2013; Fusi et al., 2016; Nieh et al., 2021). We have yet to show that the proposed approach can scale to multiple latent dimensions. Nonetheless, the current results indicate that it is possible to automatically learn continuous supported latent dimensions using standard techniques for training artificial neural networks.

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

## A  APPENDIX

### A.1  DERIVATION OF THE CONNECTIVITY MATRIX FOR THE INVERSE LAPLACE TRANSFORM

As described in Eq. 2, the inverse Laplace transform is performed using the Post inversion formula (Post, 1930):

$$\tilde{f}(\overset{*}{\tau}; t) = \mathbf{L}_k^{-1} F(s; t) = \frac{(-1)^k}{k!} s^{k+1} \frac{d^k}{ds^k} F(s; t),$$

where $\overset{*}{\tau} := k/s$. To implement this equation in a neural network we construct a discrete approximation for $\frac{d^k}{ds^k}$. First we compute $N$ by $N$ linear operator $D$ which approximates the first order derivative: $\frac{d}{ds}$ (Alg. 2) and then raise $D$ to power $k$ to implement $k$-th order derivative: $D^k \approx \frac{d^k}{ds^k}$.

---

**Algorithm 2** Constructing $N$ by $N$ linear operator $D$ which approximates the first order discrete derivative.

---

$D \leftarrow zeros(N, N)$
**for** $i \leftarrow 1$ to $N - 1$ **do**
$\quad D[i, i - 1] \leftarrow -\frac{s[i+1] - s[i]}{(s[i] - s[i-1])(s[i+1] - s[i-1])}$
$\quad D[i, i] \leftarrow \frac{\frac{s[i+1] - s[i]}{s[i] - s[i-1]} - \frac{s[i] - s[i-1]}{s[i+1] - s[i]}}{s[i+1] - s[i-1]}$
$\quad D[i, i + 1] \leftarrow \frac{s[i] - s[i-1]}{(s[i+1] - s[i])(s[i+1] - s[i-1])}$
**end for**

---

### A.2  COMPUTING COEFFICIENT OF VARIATION OF $\tilde{f}$

$\tilde{f}$ has a unimodal impulse response with peak at $t = \overset{*}{\tau}$. The coefficient of variation, $c$, is a ratio of standard deviation and mean. The mean of $\tilde{f}$ is:

$$\mu = \int_0^\infty t \tilde{f}(s; t) dt$$

$$= \int_0^\infty t \frac{1}{t} \frac{k^{k+1}}{k!} \left(\frac{t}{\overset{*}{\tau}}\right)^{k+1} e^{-k\frac{t}{\overset{*}{\tau}}} dt$$

$$= \frac{k^{k+1}}{k!} \int_0^\infty \left(\frac{t}{\overset{*}{\tau}}\right)^{k+1} e^{-k\frac{t}{\overset{*}{\tau}}} dt$$

$$= \overset{*}{\tau} \frac{k+1}{k}.$$

The standard deviation of $\tilde{f}$ is:

$$\sigma = \sqrt{\int_0^\infty (t - \mu)^2 \tilde{f}(s; t) dt}$$

$$= \sqrt{\int_0^\infty (t - \mu)^2 \frac{1}{t} \frac{k^{k+1}}{k!} \left(\frac{t}{\overset{*}{\tau}}\right)^{k+1} e^{-k\frac{t}{\overset{*}{\tau}}} dt}$$

$$= \overset{*}{\tau} \frac{\sqrt{k+1}}{k}.$$

Finally, the coefficient of variation is then:

$$c = \frac{\sigma}{\mu} = \frac{1}{\sqrt{k+1}}.$$

The coefficient of variation depends only on $k$. Therefore the variance grows with the peak time. In other words, variance is constant as a function of the logarithm of the peak time.

### A.3 VISUALIZAION OF WEIGHTS IN CNL

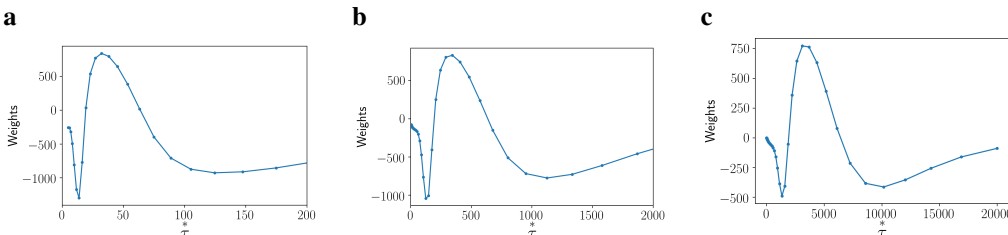

Figure 3: CNL weights for the output dense layer when $x' = 50$ (a), $x' = 500$ (b) and $x' = 5000$ (c). Since the Laplace transform is linear, the output weights are interpretable. The weights are largest around $\overset{*}{\tau} = 50$ (a), $\overset{*}{\tau} = 500$ (b) and $\overset{*}{\tau} = 5000$ (c) and have a Mexican hat connectivity pattern which helps to sharpen the prediction.

### A.4 ILLUSTRATIONS OF EXPERIMENT 2 A AND 3

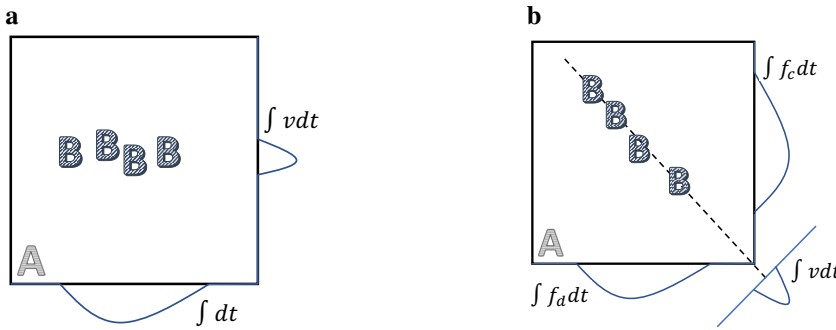

Figure 4: Illustration of predicting distance by learning latent modulatory inputs, as highlighted in Experiment 2. **a.** Let us assume that we want to estimate traveled distance from location A to location B. If we integrate time steps while traveling from A to B (x axis) then the estimate would not be accurate (broad distribution shown in blue) since our velocity changes during the trip. However, if we integrate velocity (y axis) then the estimate would be more accurate (narrow distribution shown in blue) and subject only to error in the velocity estimate itself. **b.** If velocity is not directly observable, it needs to be estimated as a combination of observable inputs, $f_c$ and $f_d$ as in Example 2a (for example, perhaps we are riding a bicycle and only observe motor outputs and sensory inputs - those would correspond to $f_c$ and $f_d$ here). If we use only $f_c$ or $f_d$, our estimate of traveled distance will not be accurate (broad distributions shown in blue). However if we find a correct combination of $f_c$ and $f_d$ (in this simple illustration that is $f_c + f_d$) then the velocity estimate is more accurate (narrow distribution shown in blue) and subject again only to error in the velocity estimate itself.

```
Target Pattern
01001000001000000110

Input Data                  S  T
00001111001001001011       0  0
11100100100100110001       0  0
01011001001001001111       0  0
01011110100000000010       0  0
01111101011111101001       1  0
11101000001001000111       0  0
11100110111001110111       0  0
10100000110000101100       0  0
11000000010010001010       0  0
01001011011100010111       0  0
01011101100000001110       0  0
01011011011011010110       0  1
10011111001111101001       0  0
01001101111110010111       0  0
```

Figure 5: Example output from the Counting dataset used in Experiment 3 (duration and size of the pattern are shortened for illustrative purposes). Input Data columns represent modulatory inputs, and rows represent time steps. In this example, the five rows highlighted in blue contain the target pattern, with the red portions being the target pattern itself. The target number of pattern occurrences is 3, corresponding to the number of patterns between the signal to start counting "S" (fed into the network as $f_a$) and the hidden target "T".

## A.5 AVERAGE DISTANCE METRIC IN EXPERIMENTS 1 AND 2A

| | x'=50 | x'=500 | x'=5000 | Params |
|---|---|---|---|---|
| CNL-F | $8.0 \pm 0.0$ | $141.0 \pm 0.0$ | $2658.6 \pm 6.1$ | **51** |
| CNL | $2.0 \pm 0.0$ | $19.0 \pm 0.0$ | $\mathbf{249.3 \pm 1.4}$ | **51** |
| RNN | $83.1 \pm 107.0$ | $641.4 \pm 612.8$ | $7513.0 \pm 10853.5$ | 4353 |
| LSTM | $11.1 \pm 41.4$ | $763.5 \pm 302.9$ | $12560.0 \pm 0.0$ | 17217 |
| GRU | $0.8 \pm 1.7$ | $102.8 \pm 405.9$ | $12405.7 \pm 154.8$ | 12929 |
| coRNN | $4.0 \pm 3.7$ | $438.3 \pm 594.5$ | $12539.0 \pm 0$ | 8385 |
| LMU | $\mathbf{0.0 \pm 0.0}$ | $\mathbf{0.0 \pm 0.0}$ | $5001.3 \pm 5724.4$ | 12610 |

Table 5: Experiment 1: The average distance between the actual timestamp of the target event and the timestamp that received the highest probability estimate.

| | x'=50 | x'=500 | x'=5000 | Params |
|---|---|---|---|---|
| CNL-F | $29.7 \pm 14.5$ | $354.0 \pm 0.0$ | $8546.1 \pm 9514.7$ | **53** |
| CNL | $\mathbf{2.4 \pm 0.7}$ | $\mathbf{35.2 \pm 5.2}$ | $\mathbf{433.4 \pm 118.1}$ | **53** |
| RNN | $57.7 \pm 22.1$ | $735.2 \pm 139.3$ | $6783.5 \pm 3448.6$ | 4481 |
| LSTM | $55.9 \pm 8.7$ | $607.1 \pm 50.6$ | $8200.2 \pm 3628.6$ | 17729 |
| GRU | $62.2 \pm 13.7$ | $592.8 \pm 90.9$ | $5293.3 \pm 1597.3$ | 13313 |
| coRNN | $49.9 \pm 10.7$ | $568.1 \pm 57.4$ | $5553.4 \pm 913.9$ | 8513 |
| LMU | $47.9 \pm 31.4$ | $778.3 \pm 128.8$ | $7884.5 \pm 2141.8$ | 12740 |

Table 6: Experiment 2a: The average distance between the actual timestamp of the target event and the timestamp that received the highest probability estimate.

A.6    VISUALIZATION OF REPRESENTATIVE EXAMPLES IN EACH EXPERIMENT

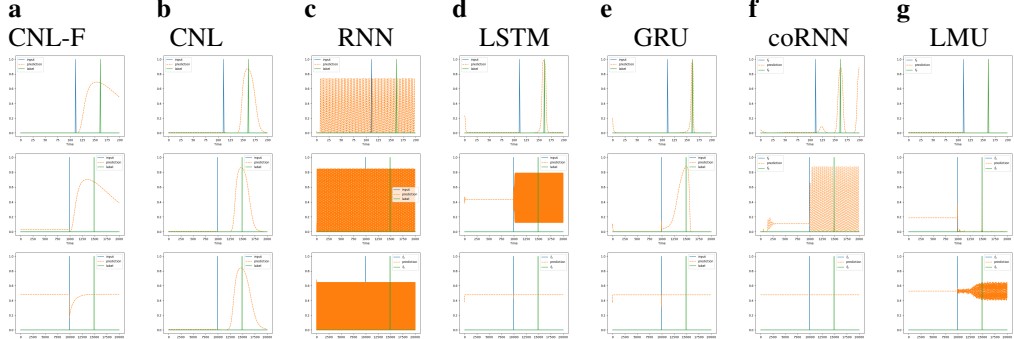

Figure 6: Experiment 1: representative examples of prediction (orange lines) for different models. Input $f_a$ is shown in blue and target $f_b$ is in green. Each row corresponds to a different time scale. Top row: $x' = 50$, middle row: $x' = 500$ and bottom row: $x' = 5000$.

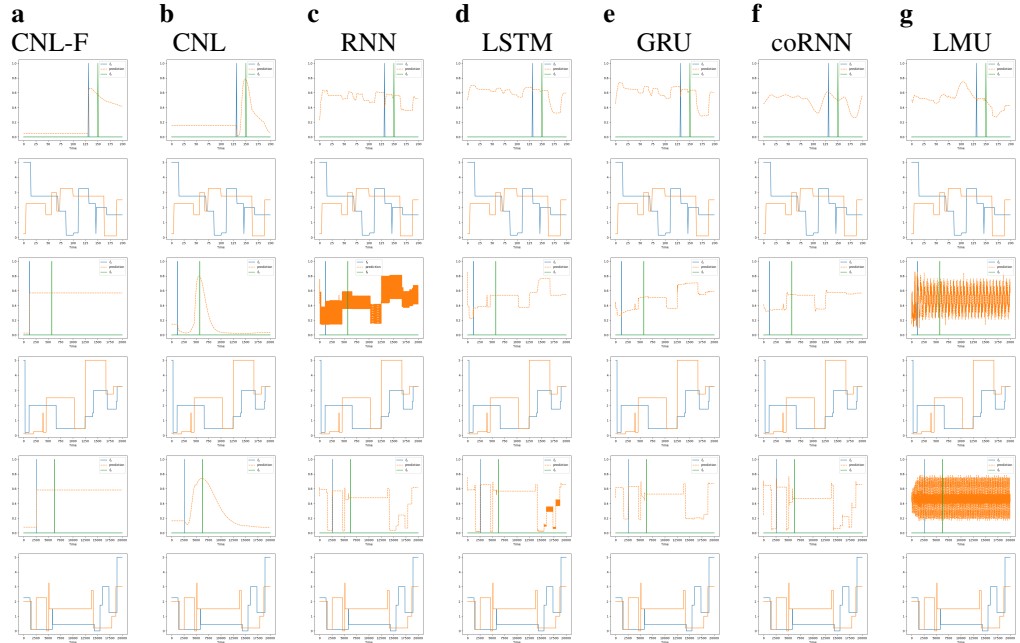

Figure 7: Experiment 2a: Odd rows: representative examples of prediction (orange lines) for different models. Input $f_a$ is shown in blue and target $f_b$ is in green. Even rows: Modulatory inputs $f_c$ and $f_d$. Plots are shown for three temporal scales. Top two rows: $x' = 50$, third and fourth row: $x' = 500$ and bottom two rows: $x' = 5000$.

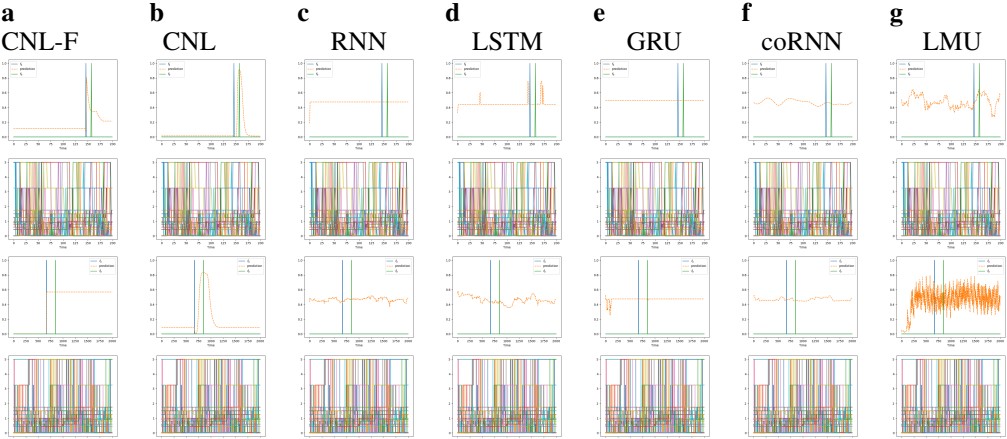

Figure 8: Experiment 2b: Odd rows: representative examples of prediction (orange lines) for different models. Input $f_a$ is shown in blue and target $f_b$ is in green. Even rows: Modulatory inputs. Plots are shown for two temporal scales. Top two rows: $x' = 50$, third and fourth row: $x' = 500$.

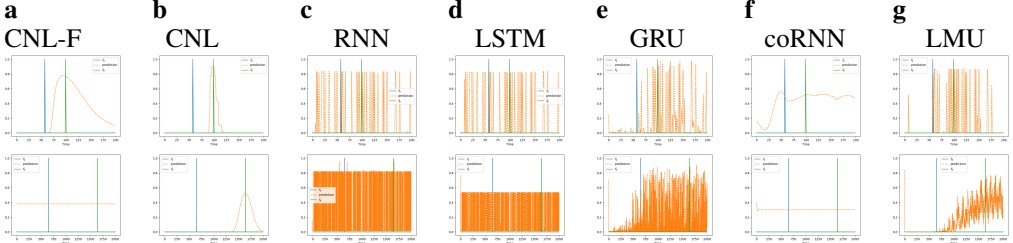

Figure 9: Experiment 3: Top row: representative examples of prediction (orange lines) for different models. Input $f_a$ is shown in blue and target $f_b$ is in green. Plots are shown for two temporal scales. Top row: $f_b = 1$ after 10 counts, bottom row: $f_b = 1$ after 200 counts.

