# OpenReview forum: "Representing Latent Dimensions Using Compressed Number Lines"
_ICLR.cc/2023/Conference — Submitted to ICLR 2023_

### Official Review · Reviewer_GVDk · 2022-10-19

**Confidence:** 4
**Correctness:** 3
**Technical Novelty And Significance:** 2
**Empirical Novelty And Significance:** 2
**Recommendation:** 1

**Clarity, Quality, Novelty And Reproducibility:**

Clarity: The paper is appropriately structured, though sometimes the motivations and the rationale of the approach are not very clear. The authors blend machine learning terminology with lexicon from the cognitive neurosciences, which might be confusing for the reader. For example, the standard notion of “Mental Number Line” does not necessarily entail temporal processing (see Dehaene, 2003) and the “number lines” mentioned by the authors seem to rather refer to the general idea that events in the environment are represented and processed spatially; indeed, except for the counting experiment the tasks discussed in the paper seem mostly related to temporal processing rather than numerical processing (also see Bonato et al., 2012). Readability of multi-panel figures should be improved.

Quality: The proposed approach is interesting and the experiments have been appropriately conducted and evaluated, though the paper would gain strength by measuring model performance on realistic data and comparing it with state-of-the-art approaches.

Novelty: The originality of the work might be limited, since it overlaps with other recently published work (DeepSITH; Jacques et al., 2021). Such overlap was emphasized in a previous version of the paper, but has been totally omitted in the current version. The proposed method has been actually renamed “Compressed Number Line (CNL)” rather than G-SITH (previous name), probably to downplay the potential overlap with DeepSITH.

Reproducibility: I think the results are reproducible.

References

Jacques, B., et al. (2021). DeepSITH: Efficient learning via decomposition of what and when across time scales. Advances in Neural Information Processing Systems.

Li, S., et al. (2019). Enhancing the locality and breaking the memory bottleneck of transformer on time series forecasting. Advances in neural information processing systems.

Dehaene, S. (2003). The neural basis of the Weber–Fechner law: a logarithmic mental number line. Trends in cognitive sciences.

Bonato, M., Zorzi, M., & Umiltà, C. (2012). When time is space: Evidence for a mental time line. Neuroscience & Biobehavioral Reviews.

Testolin, A., & McClelland, J. L. (2021). Do estimates of numerosity really adhere to Weber’s law? A reexamination of two case studies. Psychonomic Bulletin & Review.

**Strength And Weaknesses:**

One strength of the current approach is that it requires a much smaller number of parameters compared to other methods, since most of the model’s parameters are derived analytically. However, it would have been useful to test it more directly against similarly inspired work (DeepSITH by Jacques et al., 2021) and/or other state-of-the-art methods for time series analysis (e.g., LogSparse Transformer by Li et al., 2019), and to demonstrate its performance on realistic datasets, rather than just focusing on synthetic data.

Moreover, in the counting experiment, rather than just considering two extreme cases (10 vs. 200) it would be interesting to systematically study how the encoding accuracy is modulated by the number of events to be counted, to verify whether model behavior adheres to Weber’s law (see discussion in Testolin and McClelland, 2021).

**Summary Of The Paper:**

In this paper, the authors introduce a neural network model that learn to map input sequences into a logarithmically compressed temporal representation. The model is based on the analytical computation of recurrent weight matrices to construct a discrete approximation of the real part of the Laplace transform. The proposed method is tested on a set of synthetic problems, showing better performance compared to several benchmark methods.

**Summary Of The Review:**

This paper could stimulate further research work, though it seems incremental in nature. Moreover, I believe the general framework should be described and presented more clearly, and its performance should be assessed on other challenging prediction tasks and benchmarked against state-of-the-art models.

---

### Official Review · Reviewer_p2xG · 2022-10-25

**Confidence:** 3
**Correctness:** 3
**Technical Novelty And Significance:** 3
**Empirical Novelty And Significance:** 4
**Recommendation:** 5

**Clarity, Quality, Novelty And Reproducibility:**

Clarity: methods section has issues, see Weaknesses above.

Quality: concerns about the reported experimental results, see Weaknesses above.

Novelty: the proposed neural network and the experimental setup for the ML models is novel and worthwhile.

Reproducibility: while I did not thoroughly review the code, the authors appear to share the necessary code to generate the datasets, train the models, and visualize the results.

**Strength And Weaknesses:**

Strengths
- The technical contribution of recasting a Laplace transform as an RNN and the inverse Laplace transform as a dense NN is novel, to my knowledge, and can be used in other applications beyond the ones tested
- Experiment setup is novel for ML and provides challenging tasks for existing methods like LSTMs/GRUs
    - Experiment 2 is well-constructed and insightful, supporting their motivating choices for designing the CNL model. It reveals that learning temporal relationships that depend on sorting out many latent variables is fairly difficult for other networks
- The choices of alternative models to evaluate is sensible and provides good context with the existing literature on learning temporal relationships in data

Weaknesses
- I found the methods section difficult to understand from a first read. I found it easier to interpret when it laid out which variables were kept fixed and which were learnable (i.e. the last 2 paragraphs of section 2). My recommendation is to introduce this first, and then run through how you set up the NN to approximate the Laplace transform / inverse Laplace. Otherwise too many intermediate variables are introduced which do not end up being key to the final proposal.
    - for example, here's a place where this became confusing. The text says that $c$ does not depend on $\overset{\ast}{\tau}$, but based on the definitions of $c$ and the relationship between $\overset{\ast}{\tau}$ and $k$ stated after Eq. 2, it can be rewritten as $c = 1 / \sqrt{\overset{\ast}{\tau}s + 1}$. The independence from $\overset{\ast}{\tau}$ only makes sense because the experimenters are fixing $k$, $\overset{\ast}{\tau}$, and $s$.
- Some experimental choices are not clear and might have contributed to misleading results
    - Why were there different sizes of training/validation/test sets for the different experiments? No justification is given.
    - Was there any early stopping criterion? Training for 1000 epochs on so little data seems like it would result in overfitting the competing models like the LSTM.

Further suggestions
- I appreciated Tables 5 & 6, which provide support for Weber law-like behavior for the CNL on the tasks presented here. I'm curious whether this also applies to the counting task given the literature on numerosity (e.g. https://www.ncbi.nlm.nih.gov/pmc/articles/PMC7870758/). I see that the authors only tested 10 vs. 200, so testing some additional values along a log-axis may be helpful here.
- While I understand the motivation to capture existing results from the human/animal literature, this paper would be strengthened by a discussion of areas of ML that would benefit from the types of log-compressed relationships learned by the CNL. For example, do the authors envision that human-like representations of time/space as learned by the CNL would be useful in robotics or RL? Is there any utility in using CNL for learning temporal relationships in real-world time series data, especially very long temporal sequences?

**Summary Of The Paper:**

This paper first summarizes many areas of cognitive neuroscience which have shown evidence for log-compression of quantities, such as time, distance, and numerosity. This is described in psychophysics as following the Weber-Fechner law. The authors then present a method to construct a deep neural network that can learn temporal relationships between parts of a sequence by encoding continuous, log-compressed "number line" representations of latent dimensions relevant to the task. This method relies on (1) a fixed-weight recurrent matrix which implements a modified Laplace transform, (2) some learnable terms $\alpha$ which modulate the recurrent weights, (3) a dense layer that performs an approximate inverse Laplace transform, and finally (4) a trainable dense output layer. The authors then tested their proposed network on event prediction experiments that are modeled after the cognitive neuroscience experiments described at the beginning of the work. They compare the performance of their Compressed Number Line (CNL) model to other similar alternatives, such as various RNNs and the Legendre Memory Unit model. They find that their model, which has far fewer free parameters than the competing models, often fares the best.

**Summary Of The Review:**

In the current state, this work is borderline. While the paper has novel contributions that would merit a spot at the conference, I do have a couple of key concerns about the experiments that I would like for the authors to address. I am not convinced that the competing models were evaluated fairly, and this has led me to lower my correctness score and rate this paper as a marginal reject. This score can be raised with some additional details provided by the authors and/or further data/experiments to illustrate fair evaluation of the models.

---

### Official Review · Reviewer_8ytJ · 2022-10-26

**Confidence:** 4
**Correctness:** 3
**Technical Novelty And Significance:** 2
**Empirical Novelty And Significance:** 2
**Recommendation:** 3

**Clarity, Quality, Novelty And Reproducibility:**

The work is mostly clearly communicated, though the figures could be made more legible. The model is not very novel: the core “log-compressed timeline” concept comes from the existing time cell model of Laplace transform, but only added a component that extracts time-dependent latent variables which seems to be designed for the choices of experiments. I presume the work will be reproducible upon release of the code.


**Strength And Weaknesses:**

Strengths:
- The paper is clearly written with well-defined mathematical variables. I also quite like the figure of the model (Figure 1c) which is simple and clear. The paper showed the emergence of hippocampal representations such as time cells and place cells (Figure 2) while also including many experiments beyond using only time and place.

Major weaknesses:
- The CNL model doesn’t seem to deviate much from the time cell model introduced in [Liu et al., 2018](https://onlinelibrary.wiley.com/doi/10.1002/hipo.22994), except the part where they included the trainable dense network to compute  $\alpha$, the latent variable that can modulate the recurrent weights $W_L$ in the Laplace layer, from modulatory inputs. The baseline models that CNL is compared to, including RNN, LSTM, GRU, coRNN, LMU, are most often used as time cell models (wherein sequential activities can be observed in the hidden state activities), which would be fair comparisons to the “core” layers in CNL which perform Laplace and inverse Laplace transform since Liu et al showed that these core layers alone can produce time cells. In experiments 2a, 2b, and 3 (in which the latent variable $\alpha$ is involved), authors show that CNL outperforms the baseline models, which suggests the possibility that CNL relies on the feedforward layers that translate modulatory inputs to $\alpha$ to perform these tasks. What would happen if you augment these layers to the baseline models, with $\alpha$ modulating the parameters in the baseline models?
- Relatedly, since CNL performs worse than some of the baseline models in the $x’$=50 and 500 settings in experiment 1 where time is the only variable, does this mean that the feedforward layers hinder the timeline representation, at least in the $x’$=50 and 500 settings?
- In the model from Liu et al. there is a layer II that ensures Dale’s Law that this paper doesn’t include.What would happen if you consider Dale’s Law in CNL? (Also, please cite and discuss Liu et al. 2018 as this current work is largely based on that paper.)
- Comparing Table 1 and Figure 6, it seems that two models with similar binary cross-entropy loss can have very different prediction patterns (specifically I’m talking about CNL and LMU in the $x’$=500 setting). This hints at the possibility that the choice of loss function cannot capture the desired prediction pattern (a bell-shaped curve that follows Weber-Fechner law). Have you tried using other loss functions?


Minor weaknesses:
- This work only applies to latent variables that are a function of time, i.e. x(t). However, in the introduction section, the author cited a few example number line representations that are independent of time, such as numerosity in Nieder & Miller 2003 and pitch of tones in Aronov et al., 2017. Can these experimental findings involving time-independent variables be explained by CNL?

Questions:
- Section 3.3: “The 50 modulatory inputs were passed through a set of weights $W_\alpha$, which was a two layer feedforward network”. Is the feedforward network two-layer for all experiments or only this one with 50 inputs? Does the depth of the feedforward network affect CNL’s performance?
- Figure 7b first row: Why is the prediction up to the time when $f_a$ comes on a non-zero constant? This can be observed in more than one successful model (i.e. model that beat other comparison models) (another eg. would be Figure 6g second row)



**Summary Of The Paper:**

In this paper, the authors presented Compressed Number Line (CNL), a deep neural network model that can extract time-dependent latent variables from sensory inputs, and use them to make predictions through Laplace transform-based timeline representation. This method essentially generalizes the timeline model to number lines, and showed the Weber-Fechner law in the network’s prediction.

**Summary Of The Review:**

I’m leaning towards rejecting this paper for ICLR. The advances made here seem incremental when compared to Liu et al. 2018. The results are also not very strong.

---

### Official Review · Reviewer_ZzkM · 2022-11-07

**Confidence:** 2
**Correctness:** 4
**Technical Novelty And Significance:** 4
**Empirical Novelty And Significance:** Not applicable
**Recommendation:** 6

**Clarity, Quality, Novelty And Reproducibility:**

Clarity/Reproducibility: The paper is clearly written, and experiments are largely clear and reproducible with the provided details.

Quality: The results on a 3 experiment suite are convincing and their interpretation provides insights into improving the design of future sequence models.

Novelty: I am not familiar with prior work in this space and cannot comment on this paper's novelty.

**Strength And Weaknesses:**

**Strengths:** This paper is very clearly written. The suite of experiments is extensive and evaluates various aspects of the proposed method, the results of which are well discussed. The proposed approach also requires fewer parameters compared to standard sequence models such as RNNs and LSTMs.

**Weaknesses:* My main concern is the lack of a discussion on what areas of machine learning the proposed method can be applied to with significant performance gains. The experiments suggest some directions (eg. needing fewer parameters), but a concrete, consolidated discussion of on which ML tasks this method might succeed or fail is missing. It may be worth mentioning some of these tasks in the introduction too. As framed currently, the paper is largely focused on adapting neural networks to mimic a particular cognitive phenomenon, and less on arguing when and why such mimicry might be useful in practice.

**Summary Of The Paper:**

This paper proposes a method that learns to map input sequences into log-compressed number lines. The proposed method is evaluated on event prediction using synthetic data in 3 different settings and shown to outperform RNNs, LSTMs and several other baselines (despite having fewer parameters).




**Summary Of The Review:**

I think this is a well written paper with convincing results, but have not evaluated its novelty with respect to prior work. I think the paper would benefit with stronger links to the machine learning / sequence modeling literature (i.e.on what tasks would the proposed approach excel in, and on what tasks would it not)

---

### Decision · Program_Chairs · 2023-01-20

**Decision:**

Reject

**Justification For Why Not Higher Score:**

The authors did not engage in the discussion with the reviewers and in its current form the paper is not suitable for publication at a ML conference.

**Justification For Why Not Lower Score:**

NA

**Metareview: Summary, Strengths And Weaknesses:**

This paper proposes a model that can reproduce the Weber-Fechner law of representing quantities (e.g. time, distance, numerosity) in a log-compressed fashion from psychophysics research. The method uses a fixed-weight recurrent matrix with learnable modulation to map the input sequence into a log-compressed "number line" representation, and a dense layer to inverse the transformation and produce the output sequence. The authors demonstrate that their approach outperforms standard RNNs (e.g. LSTM, GRU etc) on a set of toy tasks despite having fewer parameters.

While the reviewers agreed that the paper was well written and had merit, there were questions raised about the novelty of this method, a lack of relevant citations and a lack of connection to the ML literature to make it clear what this method is useful for apart from replicating a psychophysical phenomenon. Although many reviewers were willing to increase their score had their questions been answered, the authors did not provide a response.